# Groundwater Potentiality Assessment of Ain Sefra Region in Upper Wadi Namous Basin, Algeria Using Integrated Geospatial Approaches

**Abdessamed Derdour [1], Abderrazak Bouanani [2], Noureddine Kaid [1], Kanit Mukdasai [3,\*], A. M. Algelany [4,5], Hijaz Ahmad [6], Younes Menni [1] and Houari Ameur [1]**

1. Department of Technology, University Center Salhi Ahmed Naama (Ctr. Univ. Naama), P.O. Box 66, Naama 45000, Algeria; derdour@cuniv-naama.dz (A.D.); kaid.noureddine@cuniv-naama.dz (N.K.); menni.younes@cuniv-naama.dz (Y.M.); ameur@cuniv-naama.dz (H.A.)
2. Laboratory No. 25, University of Abou Bekr Belkaid of Tlemcen, P.O. Box 119, Tlemcen 13000, Algeria; a_bouananidz@yahoo.fr
3. Department of Mathematics, Faculty of Science, Khon Kaen University, Khon Kaen 40002, Thailand
4. Department of Mathematics, College of Science and Humanities in Al-Kharj, Prince Sattam Bin Abdulaziz University, Al-Karj 11942, Saudi Arabia; ah.mohamed@psau.edu.sa
5. Department of Mathematics, Faculty of Sciences, Fayoum University, Fayoum 63514, Egypt
6. Section of Mathematics, International Telematic University Uninettuno, Corso Vittorio Emanuele II 39, 00186 Roma, Italy; hijaz555@gmail.com
\* Correspondence: kanit@kku.ac.th

**Abstract:** Water demand has been increasing considerably around the world, mostly since the start of the COVID-19 pandemic. It has caused many problems for water supply, especially in arid areas. Consequently, there is a need to assimilate lessons learned to ensure water security. In arid climates, evaluating the groundwater potential is critical, particularly because the only source of drinking water and irrigation for the community is groundwater. The objective of this report is to locate and identify probable groundwater basins in the upper Wadi Namous basin's Ain Sefra area. GIS and RS were used to evaluate the parameters of morphometry and to demarcate groundwater potential zones by using eight different influencing factors, viz., geology, rainfall, height, slope, land cover, land use, and lineaments density are all factors to consider. The analytical hierarchical process (AHP) was used to give weightages to the factors, and definitions within each attribute were sorted in order of priority for groundwater potentiality. The major findings of the research were the creation of groundwater-potential zones in the watershed. The hydrogeological zone of the basin was assessed as follows: very poor (0.56%), poor (26.41%), moderate (44.72%), good (25.22%), and very good (3.1%). The groundwater recharge potential zones are concentrated in low cretaceous locations, according to analytical data. The groundwater potential regions were checked to field inventory data from 45 water locations to corroborate the findings. The qualitative findings and the groundwater inventory data agreed 77.78%, according to the cross-validation study. The produced groundwater potential map might substantially assist in the development of long-term management plans by enabling water planners and decision-makers to identify zones appropriate for the placement of productive wells and reducing investment losses caused by well drilling failures. The results of the study will also serve as a benchmark for further research and studies, such as hydrogeological modeling.

**Keywords:** GIS; remote sensing; Ain Sefra; groundwater; sustainable management

## 1. Introduction

Groundwater has evolved as a critical source of water throughout human history to suit the needs of many sectors, including large water users such as farms, households, and corporations. Groundwater is a source of potable drinking water that is naturally filtered for effective water supply. It is among the major economic sources of potable water for both

urban and rural areas. Water sources supply drinking water to more than half of the world's population and account for 43% of all cropland. Meanwhile, 2.5 billion people throughout the world rely only on groundwater for their daily necessities [1]. In nations with rapid population growth and water shortages, decision-makers and planners face a difficult task in developing water resources. The scarcity of available potable drinking water is among the major environmental problems in the world, especially in arid areas, because of population growth, climatic conditions, and the scarcity of surface water resources. Algeria is one of the countries where per capita water availability is under the International Bank's criterion of water poorness of 1000 m$^3$/capita/year [2]. Renewable water resources in the country are estimated at 11,670 Hm$^3$/year, which corresponds to approximately 382 m$^3$/capita/year [3]. Currently, exploitable resources are estimated at just 7900 Hm$^3$/year [4]. There is more rainfall in the north of Algeria, and this made it benefit from large resources of surface water, unlike in the south of the country. The southern region depends on groundwater, especially in the southern Saharan region, where mainly fossil water is exploited, which is characterized by little recharge [5]. The water demand has been increasing considerably because of the increasing population and the proliferation of agriculture and industries, which means that some of these aquifers are being over-exploited [2,6]. Furthermore, according to the Algerian Minister of Water Resources, the quantity of drinking water consumed across the country increased in 2020 by 10% since the start of the COVID-19 pandemic, which sparked many protests, especially in Ain Sefra [7]. The increasing severity of water crises represents a real threat to sustainable development in the new millennium [8]. However, the poor knowledge of water resources, the proliferation of illicit wells, and poor coordination between the various authorities exacerbate the country's water management challenges. Water supply utilization is rising with each day in many locations throughout the world, influenced by a growing population and mishandling. In recent decades, worldwide water shortages have developed in desert and semi-arid zones. Therefore, identifying areas with groundwater potential becomes challenging [9,10].

Among the most significant procedures for the controlled use of groundwater resources is hydrogeological mapping [11,12]. Indeed, GIS and RS are considered among the best, most used tools. The fields of application of GIS are as numerous as they are varied. The combination of a geographic information system (GIS) with remote sensing (RS) has been shown to be a cost-effective, fast, and powerful technique providing useful data, such as on geology, land use, land cover, and slope, to explore and delineate of potential groundwater areas [13]. Such an integrated RS-GIS approach allows massive datasets to be used to cover vast areas, including inaccessible areas [14], thus providing a general view of huge areas for fast and cost-effective evaluation of groundwater potentiality [15]. There have been numerous uses of RS and GIS in groundwater quality investigation in latest decades. Some researchers, such as Gogu, Hallet [16], Hodlur, Prakash [17], Jaiswal, Mukherjee [18], Das and Pardeshi [19], Das, Pal [20], and Ekwok, Akpan [21] have used GIS to determinate potential zones of groundwater, while Sreedevi and Subrahmanyam [22] likewise relied exclusively on remotely sensed data to identify prospective locations of groundwater. Moreover, numerous scholars such as Fashae, Tijani [15], Bhuvaneswaran, Ganesh [23], Jasmin and Mallikarjuna [24], Shah and Lone [25], and Bhunia [26] have used GIS and remote sensing techniques together to identify potential areas of groundwater. In Algeria, water supplies depend heavily on groundwater for many purposes such as drinking, agriculture, and industry. Few researchers have addressed the problem in Algeria to evaluate the groundwater potentialities by using GIS and RS, but recently there has been a rapid rise in the use of these techniques. For example, Kebir, Bennia [27] evaluated the groundwater potential in the Tindouf Basin in the south of Algeria using GIS techniques and the index overlay model to achieve goals. Maizi, Boufekane [28] determined the potential areas of recharge using geospatial and multi-criteria decision analysis in the Macta watershed in Western Algeria after the integration of nine maps, such as lineament, land use/land cover, soil, slope, lithology, drainage, rainfall, etc., to investigate the possibility of aquifer areas in the Macta catchments. Kessar, Benkesmia [29] assessed the groundwater

potentiality zones in the region of Saida in the west of Algeria where six distinct elements, respectively lineament density, geology, elevation, land use, drainage patterns, and precipitation, have been implemented in GIS and merged with a weighted overlay to generate hydrogeological zonation maps of the region.

Like the cities located in semi-arid environments, the availability of water has always been at the heart of the concerns of local authorities in the region of Ain Sefra. It is a determining factor for the economic and social development of the region. Part of the Ksour Mountains in the Western Saharan Atlas, the region of Ain Sefra, whose history is closely linked to the water resource, constitutes a very good example for understanding the hydrogeological phenomena. Consequently, the present study's primary goal is to explore the groundwater potentiality zones in Ain Sefra catchment utilizing GIS, satellite imagery, and weighted overlay assessment based on analytic hierarchy process (AHP) methodologies. The analytic hierarchy process (AHP) derived from remote sensing and geographic information systems (GIS) is deemed a fundamental, efficient, consistent, and cost-efficient strategy [30–34].

*Study Area*

Ain Sefra is in the southwest of Algeria, in the west part of the Ksour Mountains. The geographic situation is between 3,580,000 to 3,660,000 m north latitude and 680,020 to 780,050 m east longitude (Figure 1). The watershed spans a total surface area of 4590.2 km², or about 15.4% of the Wilaya of Naâma's entire area. The research area's estimated elevation ranges between 804.5 m and 2219.4 m. The study area represents the upper side of the large hydrological Saharan system of Wadi Namous. Due to its geographic location, Ain Sefra shows arid to semi-arid climatic conditions, and the hydrological behaviors are characterized by spatial and temporal variability [35]. From a hydrogeological point of view, the watershed is characterized by three aquifer systems of groundwater resources: the quaternary aquifer, the aquifer of lower cretaceous sandstones, and that of the Jurassic sandstones [36]. The region's groundwater resources are extracted through wells, deep wells, and springs. As per the 2008 census, the region of Ain Sefra, which includes the municipalities of Ain Sefra, Sfissifa, Tiout, and Moghrar, had a total population of 65,860 inhabitants [37]. To supply drinking water and irrigation, the population in the study area is primarily reliant on groundwater sources. Mineral soils, limestone-magnesia soils, saline soils, and poorly graded soils are the four types of soils found in the Ain Sefra basin [38]. Except in the mountains, where remnants of primeval forests can be found, the vegetation of the area is characterized by a steppe physiognomy. From the south to the north, the aspect of the steppe changes with the composition of the soil and the rainfall gradient. The aspect of the steppe changes with the rainfall gradient and the nature of the soil from the south to the north. It should be emphasized that the Ain Sefra region still has a major sand encroachment problem in many of its communities [39].

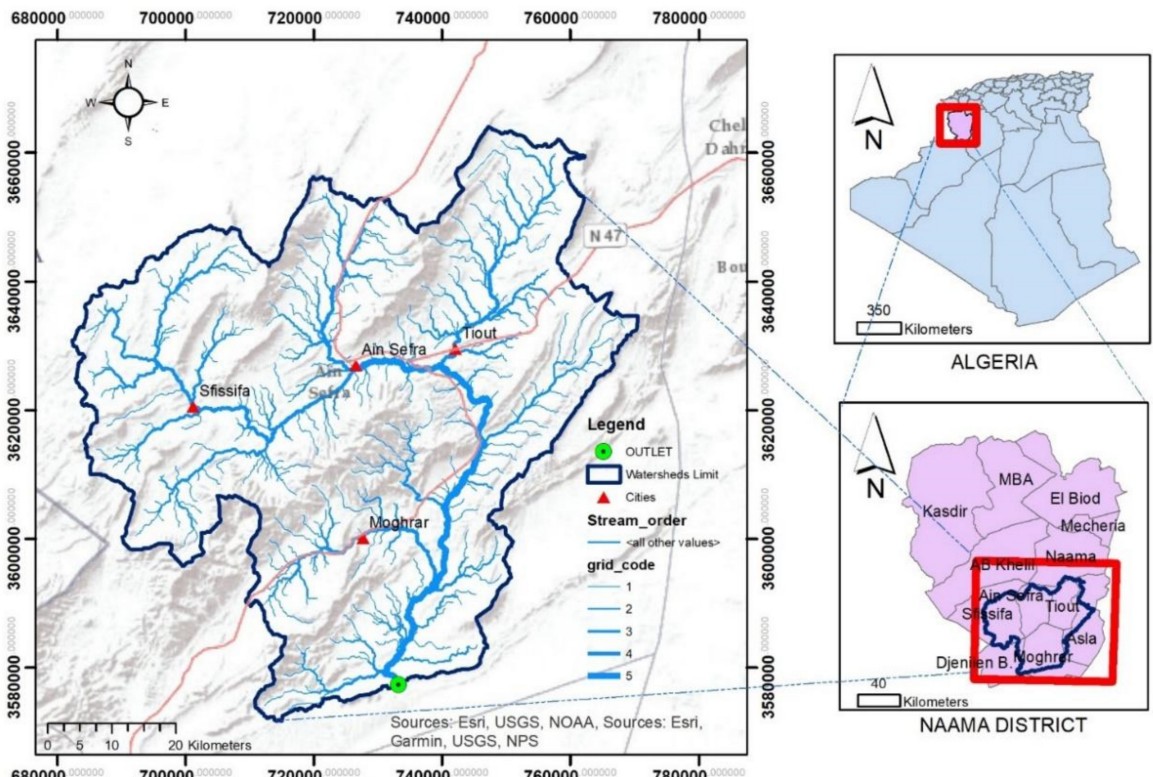

**Figure 1.** Study area.

## 2. Materials and Methods

### 2.1. Data Collection

Potential groundwater regions in the catchment of Ain Sefra were assessed using a variety of potential datasets such as geology, elevation, rainfall, topographic wetness index (TWI), land use, slope, drainage density (DD), and density of lineament (LD). In the end, we performed an additional yield data of 45 groundwater samples representing the region's aquifers to validate our results. The processes adopted in this analysis for demarcating various groundwater prospect zones in the watershed of Ain Sefra have been portrayed in Figure 2.

The research area's digital elevation model (DEM) was retrieved from the website of Earth Data with a 30 m resolution [40]. It has been used in ArcGIS software to generate the elevation map, to delineate the watershed boundaries, to derive the stream networks, and to prepare the slope map. GIS techniques were utilized to analyze various morphometric parameters for the study area, including drainage pattern, relief characteristics, drainage texture, and basin geometry.

The Topographic Wetness Index (TWI) is a frequently used tool for evaluating the influence of topography on potential soil wetness by assessing infiltration conditions using slope [41,42]. By using the slope generated previously and the flow accumulation, the TWI was calculated, following the Equation (1):

$$TWI = (As/tan\beta) \tag{1}$$

where $\beta$ is the gradient of the slope, as well as the rising slope's cumulative surface.

The region of Ain Sefra has only one ground weather station, located at the city of Ain Sefra. Therefore, spatial data of rainfall were acquired from the website of the Climatic Research Unit (CRU) to cover all of the watershed [43]. The data obtained were then interpolated spatially using the Kriging method to obtain the entire study area rainfall distribution map.

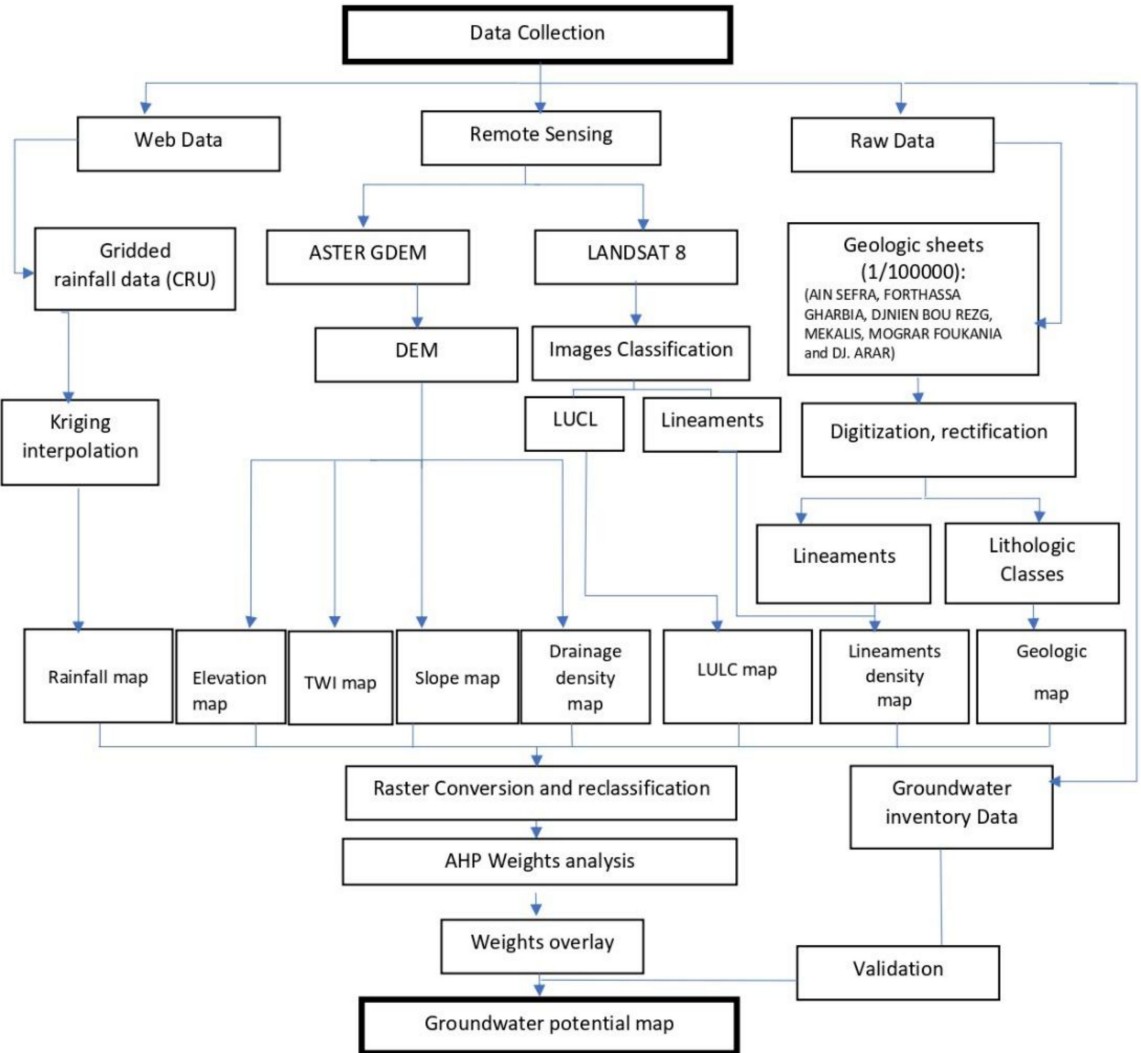

**Figure 2.** Methodology employed in the present study.

The Landsat satellite images with 30 m spatial resolution and 11 spectral bands were acquired free of charge from the Earth Data website to generate the land use and land cover map (LUCL) of the Ain Sefra watershed [44]. The LULC map was extracted from a mosaic of four Landsat images through supervised classification by using the ArcGIS environment.

The nature of the geological rocks outcropping on the surface directly influences the groundwater recharge [45]. The geological maps used in this study were obtained from the Algerian Geological Service Agency (AGSA) at a scale of 1:100,000. Six (06) geological map sheets were scanned, georeferenced, digitized, and merged to obtain the geological map of the whole study area.

Lineaments of an area represent rock fractures, joints, and faults, which act as pathways of groundwater recharge [46]. They were digitized from the geological map and also automatically extracted by using the PCI Geomatica software.

The density of drainage (DD), known as the catchment area, is defined as a natural unit draining the surface runoff into a specific unique point [47]. Using a line density analysis tool, the density of drainage and the density of lineament maps were generated using the clipped DEM.

The groundwater inventory data were gathered from a variety of trustworthy sources such as the department of water resources of the Wilaya of Naâma, and the high commission for the development of rangelands (HCDS). Table 1 shows the data used in this study, its source and its purpose.

**Table 1.** Types of data collected.

| No | Data Collected | Source of Data | Generated Variable Layer |
|---|---|---|---|
| 1 | ASTER DEM (30 m resolution) | Accessed on 6 June 2020 from https://earthexplorer.usgs.gov/ | DEM map Density of drainage map Slope map Topographic Wetness Index map |
| 2 | Grid-based rainfall data | Accessed on 6 June 2020 from Climatic Research Unit, website: https://sites.uea.ac.uk/cru/data | Areal rainfall map |
| 3 | Landsat satellite images (Landsat 8 OLI and TIRS Level-1 Data Products) | Accessed on 11 June 2020 from https://earthexplorer.usgs.gov/ | LULC map Lineament density map |
| 4 | Existing maps | Algerian Geological Service Agency. Department of Water Resources of Wilaya of Naâma. | Geological map |
| 5 | Existing wells/springs | High Commission for the Development of rangelands, Naâma. | Groundwater inventory data map. |

### 2.2. Assigning Rank and Weight

For this study, eight potential layers were considered: geology, density of drainage, slope, rainfall, LULC, topographic wetness index, density of lineaments, and elevation. The thematic maps were classified and converted into raster layers and were displayed into UTM Projection Datum WGS 84, Zone 30. Then, the groundwater potential maps were acquired by superimposing all the thematic layers, depending on the weightage overlay technique.

The analytical hierarchy process (AHP) was employed in this study to identify the most influencing parameters. The AHP technique is based on the hierarchical organization of decision criteria in decision-making problems, and it has become one of the most important tools in the field of decision making across numerous disciplines [33,48–51]. Based on the AHP method, a pairwise comparisons matrix was used to assign weights by using the flexible AHP spreadsheet template developed by Goepel [52]. By rating each element based on the pairwise comparison, each factor was awarded a rank in a range of one (equal significance) to nine (extreme significance), according to the Saaty's scale [53].

The parameter with a high weight represents a layer with high impact on groundwater potential, and the minor impact on groundwater potential is related to the parameters with low weight. This method assesses the significance of a parameter that affects groundwater potential [51]. Table 2 shows Saaty's classification scale, according to [53]. Many authors' previous experiences in geology, hydrogeologic, and geospatial research support the knowledge-based strategy for groundwater potential indexing employed in this study [54,55], coupled with the opinion of experts and specialists in the study area [36,39,56]. The highest weights were given to geology and drainage density, while rainfall, slope, and altitudes were provided moderate weights, and density of lineament and land use land cover LULC were attributed low weights.

**Table 2.** Saaty's classification scale.

| AHP Scale | Description |
|---|---|
| 1 | Equal |
| 3 | Moderate |
| 5 | Strong |
| 7 | Very strong |
| 9 | Extreme |

2,4,6,8 can be employed to describe values that are in the middle.

### 2.3. Groundwater Potential Zones Delineation

Using the weighted overlay tool in ArcGIS software, the eight thematic layers and their respective proportion effects were combined to produce a map of potential regions for groundwater in the watershed of Ain Sefra, as shown by Equation (2):

$$GWPZ = \begin{aligned} &G_W G_{W_i} + D_W D_{W_i} + R_W R_{W_i} + T_W T_{W_i} + A_W A_{W_i} + S_W S_{W_i} + \\ &L_{UW} L_{UW_i} + L_{DW} L_{DW_i} \end{aligned} \tag{2}$$

where $G$: geology, $D$: density of drainage, $R$: rainfall, $T$: TWI, $A$: altitudes, $S$: slope, $L_U$: LULC, $L_D$: density of lineament, $W_i$: normalized weight, and $W$: normalized weight of a theme.

The model performed first the weights assignment to every thematic map using the reclassify and raster calculator tools of the spatial analyst tools in ArcGIS. The thematic features were then overlaid using the raster calculator tool of the same toolset. Afterward, to create the final groundwater potential zones, the potential zones of groundwater were categorized and divided into five different zones called very low, low, moderate, good, and excellent.

### 2.4. Validation Process

Validation is a key phase in every model's evaluation process. Many researchers used existing wells data at various locations to validate groundwater potential zones [22,47,49,57,58]. The groundwater potential zones created using the GIS and RS approaches were compared to existing groundwater yield data in the research area to evaluate the veracity of our findings. This would build more confidence for validating our results. For this purpose, we employ survey yield data collected from 41 wells and four springs existing in our region to validate the potential zones of groundwater; the data of the validation are presented in Table 3. The unpublished dataset was collected from the high commission for the development of rangelands of Naâma and the department of water resources of the Wilaya of Naâma (DRE), to obtain a larger overview of the groundwater potentiality in the watershed of Ain Sefra. If the majority (more than 50%) of groundwater inventory data agrees with the corresponding groundwater potential zone classifications, the final groundwater potential zone maps are considered valid.

**Table 3.** Yield data used for validation of groundwater potential zones.

| N° | Water Points | Names | X (UTM) | Y (UTM) | Yield (L/s) |
|---|---|---|---|---|---|
| 1 | Skhouna | W1 | 721,352.44 | 3,621,944.54 | 17.00 |
| 2 | Slih | W2 | 734,714.32 | 3,628,682.85 | 85.00 |
| 3 | Mehisserat 1 | W3 | 732,853.98 | 3,627,880.30 | 13.00 |
| 4 | Mouillah | W4 | 723,522.18 | 3,626,270.89 | 30.00 |
| 5 | Tirkount 2 | W5 | 723,271.12 | 3,634,574.08 | 8.00 |
| 6 | Ain 'Tirkount | W6 | 723,271.12 | 3,634,574.08 | 6.00 |
| 7 | Tirkount 1 | W7 | 716,957.36 | 3,636,435.03 | 7.00 |
| 8 | Hammar | W8 | 722,633.88 | 3,626,022.21 | 6.00 |
| 9 | Mehisserat 2 | W9 | 731,849.42 | 3,627,925.91 | 12.00 |
| 10 | Skhouna 2 | W10 | 721,136.34 | 3,622,233.40 | 20.00 |
| 11 | Slih 2 | W11 | 733,053.80 | 3,627,891.15 | 13.00 |
| 12 | Hopital1 | W12 | 725,522.94 | 3,626,700.28 | 5.00 |
| 13 | Mekalis | W13 | 736,075.26 | 3,651,003.74 | 2.00 |
| 14 | Ain-Sefra F2 | W14 | 718,005.92 | 3,628,560.22 | 22.00 |
| 15 | Ain-Sefra F3 | W15 | 716,552.69 | 3,627,294.86 | 28.00 |
| 16 | Benhandjir 3 | W16 | 713,111.52 | 3,622,398.54 | 6.00 |
| 17 | skhouna 3 | W17 | 720,685.00 | 3,621,734.00 | 10.00 |
| 18 | Matlag | W18 | 743,023.65 | 3,628,833.75 | 20.00 |
| 19 | Tiout 2 | W19 | 746,359.00 | 3,628,313.47 | 8.00 |
| 20 | Maader 1 | W20 | 738,462.21 | 3,629,788.43 | 18.00 |

**Table 3.** *Cont.*

| N° | Water Points | Names | X (UTM) | Y (UTM) | Yield (L/s) |
|---|---|---|---|---|---|
| 21 | Maader 2 | W21 | 740,980.75 | 3,633,232.33 | 18.00 |
| 22 | Maader 3 | W22 | 738,682.65 | 3,633,107.39 | 15.00 |
| 23 | Sam | W23 | 752,346.08 | 3,617,714.92 | 14.00 |
| 24 | Maader 4 | W24 | 743,972.88 | 3,633,495.20 | 30.00 |
| 25 | Sfissifa 2 | W25 | 700,346.35 | 3,621,303.96 | 7.00 |
| 26 | Sfissifa 3 | W26 | 700,288.75 | 3,624,206.33 | 15.00 |
| 27 | Benhandjir 1 | W27 | 711,306.72 | 3,616,889.95 | 14.00 |
| 28 | Mekhizene | W28 | 702,474.37 | 3,628,232.63 | 5.00 |
| 29 | Sfissifa 4 | W29 | 703,581.00 | 3,620,778.45 | 5.00 |
| 30 | Sfissifa 5 | W30 | 704,358.00 | 3,620,152.00 | 8.00 |
| 31 | Belgoured | W31 | 709,156.00 | 3,619,537.00 | 30.00 |
| 32 | Moghrar 4 | W32 | 726,957.20 | 3,599,904.19 | 15.00 |
| 33 | Moghrar Fougani | W33 | 728,901.16 | 3,601,011.54 | 4.00 |
| 34 | Nessissa | W34 | 741,913.67 | 3,583,679.74 | 12.00 |
| 35 | Zaouche 1 | W35 | 733,900.54 | 3,588,256.34 | 13.00 |
| 36 | Zaouche 2 | W36 | 732,955.54 | 3,587,203.22 | 13.00 |
| 37 | Draa Saa | W37 | 736,305.22 | 3,601,212.37 | 7.00 |
| 38 | Moghrar Fougani | W38 | 728,439.54 | 3,600,285.12 | 12.00 |
| 39 | Sidi Brahim | W39 | 722,422.74 | 3,583,727.58 | 4.00 |
| 40 | Oglat 1 | W40 | 718,830.68 | 3,594,554.22 | 15.00 |
| 41 | Oglat 2 | W41 | 717,051.19 | 3,595,960.67 | 14.00 |
| 42 | Rhouiba | W42 | 741,280.86 | 3,609,371.01 | 2.50 |
| 43 | Ain Aissa | S1 | 736,155.00 | 3,644,892.00 | 2.00 |
| 44 | Ain El Beida | S2 | 729,863.00 | 3,634,970.00 | 2.00 |
| 45 | Ain Srera | S3 | 732,319.00 | 3,625,181.00 | 2.00 |
| 46 | Biri | S4 | 708,839.00 | 3,640,412.00 | 2.00 |

## 3. Results and Discussion

This section contains the illustration of the results of the main impacting elements: density of drainage, geology, rainfall, elevation, TWI, LULC, and density of lineaments. In addition, it contains the final map of groundwater potentialities of the watershed of Ain Sefra. Based on the findings of this investigation, a full discussion follows:

### 3.1. Geology

The percolation and infiltration of surface water into aquifers are heavily influenced by geology. Our region is part of the Ksour Mountains. Therefore, its geological background has been studied extensively in the past [59–62]. Figure 3a shows the geology map obtained from the watershed of Ain Sefra. The geological age of the study area ranges from the Triassic to the Quaternary, with Mesozoic formations dominating, consisting of sandstone rocks units [35]. The sandstones generally occupy the center of the synclines of the Ksours Mountains. They are present in large successive banks. Jurassic sandstones are widespread over 2022.7 km$^2$, covering about 44.05% of the study area; their permeabilities are generally flat. The Quaternary covers about 1112.68 km$^2$ (24.23%) of the area. It consists of alluvium, sand, and gravel, known by their high infiltration. Low Cretaceous sandstones covering about 627.17 km$^2$ (13.66%) of the area; their permeabilities are in general high [36]. Tertiary limestones represent about 785 km$^2$ (17.11%) of the study area, known by their medium permeability. The other (<2%) is filled with Triassic and Berriasian formations.

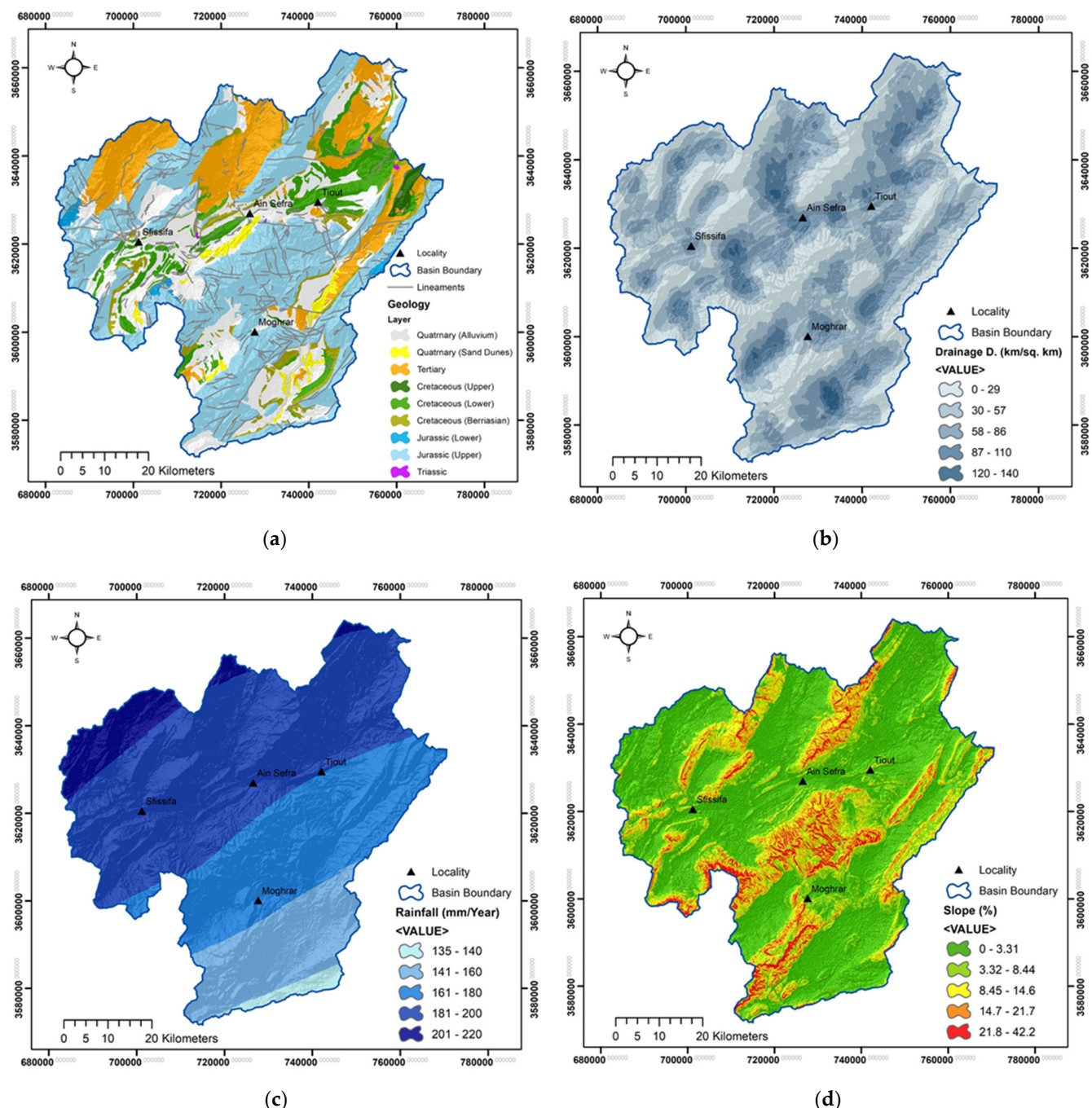

**Figure 3.** *Cont.*

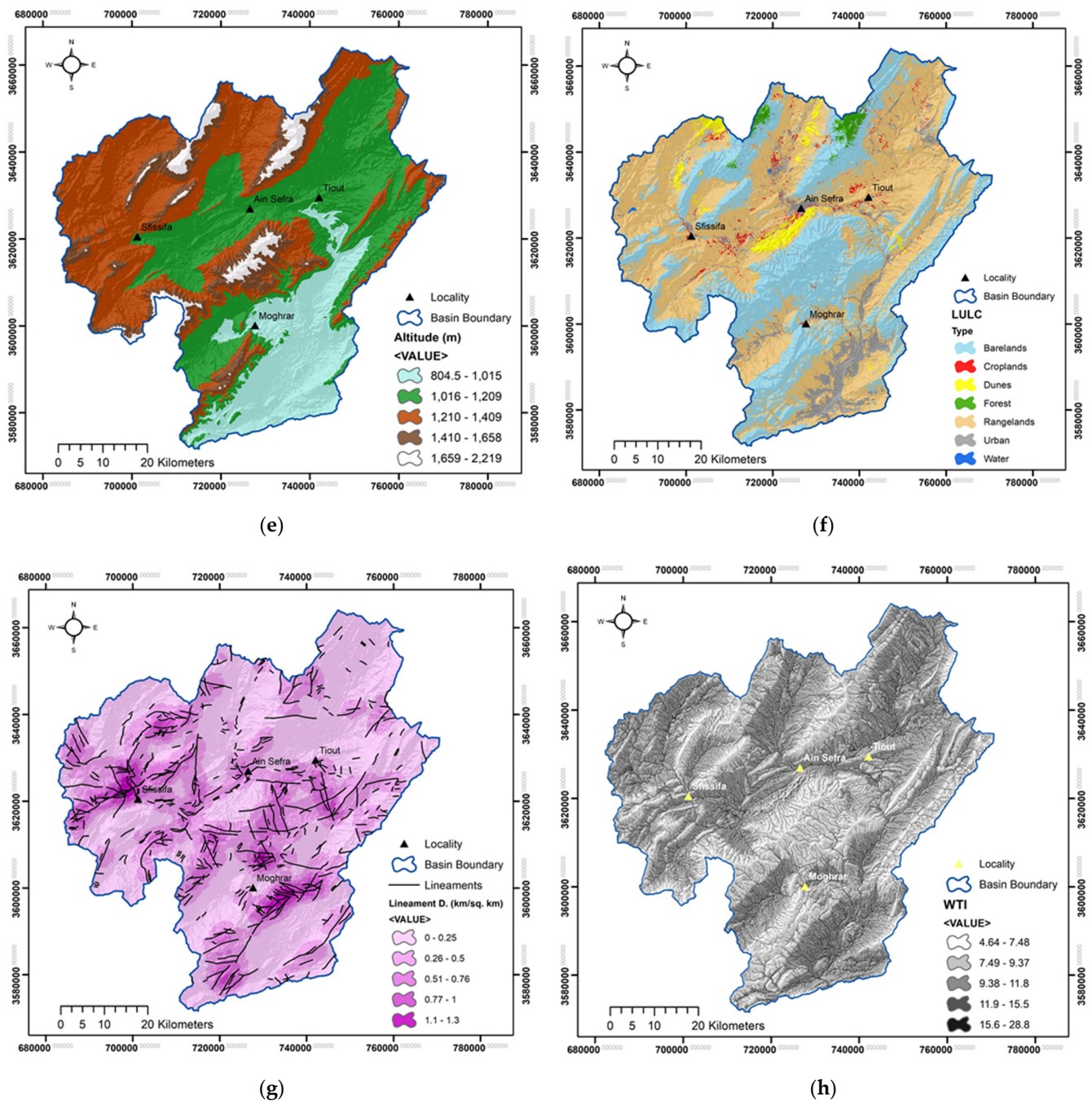

**Figure 3.** Influencing factors maps of Ain Sefra watershed. (**a**) Geology map. (**b**) Drainage density.
(**c**) Rainfall map. (**d**) Slope Map. (**e**) Altitude Map. (**f**) LULC map. (**g**) Lineament density map.
(**h**) Topographic Wetness Index map.

### 3.2. Drainage Density

The drainage density is one of the most important parameters influencing the transportation and recharging of water into the soil [63]. The drainage density is calculated by multiplying the total length of all streams (wadis) in a watershed by the entire area of the watershed. The structural analysis of a drainage network aids in determining the characteristics of a groundwater recharge zone [30]. The drainage density is influenced by many factors such as geology, topographical features of the watershed, and, to some extent, climatic and anthropogenic factors. The drainage density depends on the topographical characteristics, the geology, and the anthropogenic and climatic conditions. Drainage

density data were calculated as the flow length per unit and it was found varying from very low to very high. In the Ain Sefra watershed, most of the drainage originates from mountain peaks, and the drainage pattern is generally dendritic. Figure 3b illustrates the drainage density of the study area. As shown in the figure, the high values of drainage density are found in the flow zones of the wadis, which are composed of recent alluvial sediments. In contrast, mountainous areas are characterized by very low drainage density. The high drainage textures suggest extremely porous and permeable rock formations, while in the opposite, the fine drainage structures are more common in fractured rock formations.

### 3.3. Rainfall

The unequal distribution of precipitation in the world implies an uneven distribution of surface and groundwater. Rainfall is the primary source of groundwater recharge in arid zones, since water infiltrates underground through fractures and pores, while groundwater is accessed by shallow wells and open dug wells [64]. The map was generated using the Kriging method. The region's rainfall was classified into five classes, each with a 20 mm interval (Figure 3c). The distribution of precipitation in the study area varies from 134.6 to 205.23 mm, suggesting an arid climate. Groundwater potential is high in areas with more rainfall, which spans about 313.87 km$^2$ (6.84%), while areas suggesting low groundwater potential represent approximately 77.49 km$^2$ (1.7%). A closer look at the thematic precipitation map reveals that the northern western part had relatively high precipitation. However, the southern part received less precipitation.

### 3.4. Slope

The slope represents an important thematic layer for groundwater occurrence and recharge. The term "slope" refers to regional and local relief that has a substantial impact on aquifer recharge and groundwater potentiality [65]. Regions of gentle slopes collect rainwater and allow it to infiltrate into the soil, eventually recharging the underlying aquifers. Whereas, for higher slopes, the opposite is true [66]. In our study area five types of slopes shown in Figure 3d, viz., very low slopes between 0 and 3.3% where the presence of a relatively flat valley, low slopes between 3.3 and 8.44%, moderate slopes between 8.45 and 14.6%, relative steep slopes between 14.7 and 21.7% where the presence of pediments and structural hills, and steep slopes between 21.8 and 42.2% explained by the presence of mountains. In the low slope area, surface runoff is low, providing more time for rainfall penetration, whereas in the high slope area, high runoff is enhanced with a short residence time for infiltration and recharge. Consequently, based on a slope's influence, 59.66% of slopes in the Ain Sefra watershed were best rated in terms of potentiality of groundwater.

### 3.5. Elevation and Morphometric Parameters

The altitudinal distribution in the study area was delineated using ASTER GDEM data. The elevation map in Figure 3e indicates a range of 804.5 m to 2219.4 m. The higher altitude was represented by the peak of Djebel Aissa in the northern part, while the lower elevation was found to the south, representing the outlet of the watershed. A total of 33.9% of the study area had an elevation of less than 1200 m. Because of the virtually flat terrain, runoff will be gradual in low-elevation locations, giving rainwater more time to infiltrate. Conversely, in high elevation where slopes are higher, the runoff is relatively high and less likely to be available to groundwater. In general, on the south side of the watershed of Ain Sefra, the values reflect the character of a plain; on the other hand, in the north, the importance of high and medium altitudes is well illustrated, as this characteristic promotes runoff. Based on DEM, hydrologic formulas, and GIS techniques, various morphometric parameters in the watershed of Ain Sefra were utilized to determine relief characteristics, basin geometry, and drainage pattern. Overall, 30 quantitative morphometric parameters were estimated. Table 4 shows the morphometric parameters of the watershed of Ain Sefra, as well as the methodology used to derive them.

**Table 4.** Morphometric parameters of the watershed of Ain Sefra.

| | Morphometric Parameters | Formula | References | Results |
|---|---|---|---|---|
| **A. Basin geometry** | | | | |
| 1 | Area (A) | GIS analysis/DEM | [67] | 4590.20 km$^2$ |
| 2 | Perimeter (P) | GIS analysis/DEM | [67] | 410.95 km |
| 3 | Length (Lb) | GIS analysis/DEM | [67] | 168.92 km |
| 4 | Mean Basin Width (Wb) | Wb= A/Lb | [68] | 11.16 km |
| 5 | Constant of Channel Maintenance (C) | C = 1/Dd | [67] | 2.27 km/km$^2$ |
| 6 | Form factor (Ff) | Ff = A/Lb$^2$ | [68] | 0.16 |
| 7 | Elongation ratio (Re) | Re = 1.128 $\sqrt{A}$/L | [68] | 0.45 |
| 8 | Circularity ratio (Rc) | Rc = 4$\pi$A/P$^2$ | [69] | 0.34 |
| 9 | Compactness coefficient (Cc) | Cc = 0.2841·P/A$^{0.5}$ | [70] | 1.72 |
| **B. Drainage network** | | | | |
| 10 | Stream order (So) | Hierarchical rank | [69] | 1–5 |
| 11 | Stream Number (Nu) | Nu = N1 + N2 + … + Nn | [68] | 1510 |
| 12 | Stream Length (Lu) | Lu = L1 + L2 + … + Ln | [69] | 2016.7 km |
| 13 | Mean Stream Length (Lum) | Lsm = Lu/Nu | [68] | 2.35 |
| 14 | Mean Stream Length Ratio (Lurm) | Lurm = Lu/Lu−1 | [68] | 0.91 |
| 15 | Weighted Mean Stream Length Ratio (Lurwm) | | [68] | 0.86 |
| 16 | Bifurcation Ratio (Rb) | Rb = Nu/Nu+1 | [69] | 1.75 |
| 17 | Rho Coefficient ($\rho$) | $\rho$ = Lur/Rb | [68] | 0.51 |
| 18 | Drainage Texture (Dt) | Dt = Nu/P | [68] | 1.88 |
| 19 | Stream Frequency (Fs) | Fs = Nu/A | [68] | 0.16/km$^2$ |
| 20 | Drainage density (Dd) | Dd = Lu/A | [68] | 0.43 km/km$^2$ |
| **C. Relief characteristics** | | | | |
| 21 | The maximum height | GIS analysis/DEM | - | 2219.4 m |
| 22 | The minimum height | GIS analysis/DEM | - | 804.5 m |
| 23 | Basin relief (R) | R = H − h | [69] | 1414.9 m |
| 24 | Relief ratio (Rhl) | Rhl = H/Lb | [67] | 8.37 |
| 25 | Relative relief ratio, (Rhp) | Rhp = H.100/P | [71] | 275 |
| 26 | Ruggedness Number (Rn) | Rn = Dd.(H/1000) | [72] | 0.60 |
| 27 | Total contour length (Ctl) | GIS analysis/DEM | - | 5182.2 km |
| 28 | Slope analysis (Sa) | GIS analysis/DEM | [73] | 0–42.2° |
| 29 | Average slope (S) % | S = (Z × (Ctl/H))/(10.A) | [73] | 0.17% |
| 30 | Mean Slope of Overall Basin ($\Theta$s) | $\Theta$s = (Ctl . Cin)/A | [74] | 0.11% |

*3.6. Land Use and Land Cover (LULC)*

LULC plays a crucial role in groundwater recharge through runoff and infiltration in any region in the world. Pastoral lands, as indicated in Figure 3f, span an area of about 2715.82 km$^2$, accounting for about 59.17% of the total area, followed by bare lands, which comprise 1635.12 km$^2$ (35.62%). The dunes comprise an area of 107.95 km$^2$, or about 2.35% of the total study area. Agriculture and forest areas constitute only 1.72% and 1.05% of the study area, respectively, while about only 4.31 km$^2$ (0.09%) are covered by water bodies, located in the artificial hill reservoir in the region of Sfissifa.

*3.7. Lineament Density*

Lineaments are the manifestation of deeper geological structures at the Earth's surface. It has been used in many fields such as the study of geothermal resources, geological disasters and earthquakes, the investigation of mineral distribution, the assessment of groundwater potential, and the determination of runoff water harvesting potential areas. The interpretation of the lineament for hydrogeological purposes is based on the idea that a much higher intensity of lineaments likely results in faulty areas of groundwater conductivity. Areas of very high lineament density constitute the favorable pole for the existence of a potential aquifer, on the other hand, areas of zero or very low-density bode very unfavorably for the presence of an aquifer. The study area is crisscrossed with lineaments that have resulted from several past tectonic activities, as illustrated by the lineaments density map. The main direction is oriented in SW–NE.

*3.8. TWI*

The TWI has a big impact on how surface runoff moves and accumulates on the ground. Figure 3h shows the distribution of the topographic wetness index for the watershed of Ain Sefra. The values of TWI vary between 4.64 to 28.8. The high values are related to the low altitudes, whereas the low values of TWI are situated in the high altitude's zones. The lowest rank was given to the very low TWI values, while the highest rank was given to the extremely high TWI values, which indicates a tendency for soil moisture accumulation zone.

*3.9. Analytical Hierarchical Process Results*

The targeted objective of the AHP was the delineation of the groundwater potential zones. Eight thematic maps associated with their specific subclasses were utilized and given weight for a better achievement of the objective according to their specific influences to groundwater prospects. The geology theme was found to be the most important with the weight of 36.8% followed by the drainage density with the weight of 18.4%, then rainfall with the weight of 12.25%. The topographic wetness index was given a weight of 9.2%. It was followed by the slope with 7.35%. The land use and land cover, and the altitudes were given the weights of 6.1% and 5.3%, respectively. The lowest weight of 4.6% was assigned to the lineament density. Individual ranks for sub-variables were assigned once weights were assigned to the appropriate parameters. The highest groundwater potentiality was defined by the maximum value, and vice versa. Table 5 illustrates the ranks and the weights assigned of influencing factors of the study area.

**Table 5.** Weights are ranks assigned to influencing factors for the watershed of Ain Sefra.

| Factors | Subclasses | Influence (%) | Rank | Area (km$^2$) |
|---|---|---|---|---|
| Geology | Quaternary | 36.80 | 5 | 1483.28 |
| | Tertiary | | 3 | 785.64 |
| | Cretaceous (Upper) | | 3 | 38.00 |
| | Cretaceous (Friable sandstones) | | 4 | 38.41 |
| | Cretaceous (Sandstones with clays) | | 3 | 218.59 |
| | Upper Jurassic (Hard sandstones) | | 2 | 1920.58 |
| | Lower Jurassic (Hard sandstones) | | 2 | 102.13 |
| | Trias (Clay, Gypsum) | | 1 | 3.57 |
| Drainage density | Very low (0–29) | 18.40 | 1 | 1042.71 |
| | Low (30–57) | | 2 | 1136.49 |
| | Moderate (58–86) | | 3 | 1121.26 |
| | High (87–110) | | 4 | 917.86 |
| | Very high (120–140) | | 5 | 371.88 |
| Rainfall | 135–139 mm | 12.25 | 1 | 77.77 |
| | 140–159 mm | | 2 | 651.15 |
| | 160–179 mm | | 3 | 1270.08 |
| | 180–199 mm | | 4 | 2277.72 |
| | 200–220 mm | | 5 | 313.48 |
| TWI | 0–7.47 | 9.20 | 1 | 1247.1689 |
| | 7.47–9.36 | | 2 | 1906.5952 |
| | 9.36–11.82 | | 3 | 931.86453 |
| | 11.82–15.51 | | 4 | 389.59978 |
| | 15.52–28.76 | | 5 | 114.97162 |
| Slope | Very low slope (0–3.31%) | 7.35 | 5 | 2723.68 |
| | Low slope (3.32–8.44%) | | 4 | 756.3 |
| | Moderate slope (8.45–14.6%) | | 3 | 560.92 |
| | Relative steep slope (14.7–21.7%) | | 2 | 378.45 |
| | Steep slope (21.8–42.2%) | | 1 | 170.85 |

**Table 5.** *Cont.*

| Factors | Subclasses | Influence (%) | Rank | Area (km$^2$) |
|---|---|---|---|---|
| LULC | Water bodies | 6.10 | 5 | 3.94 |
| | Pastoral lands | | 4 | 2438.85 |
| | Forest lands | | 1 | 48.72 |
| | Dunes | | 3 | 106.47 |
| | Bare lands | | 2 | 1934.54 |
| | Agricultural lands | | 3 | 57.68 |
| Altitude | 804–1015 m | 5.30 | 5 | 791.1 |
| | 1015–1209 m | | 4 | 1579.25 |
| | 1209–1409 m | | 3 | 1561.88 |
| | 1409–1658 m | | 2 | 466.43 |
| | 1658–2219 m | | 1 | 191.54 |
| Lineament density | 0.26 | 4.60 | 1 | 1345.768 |
| | 0.27–0.53 | | 2 | 1388.146 |
| | 0.54–0.79 | | 3 | 1085.723 |
| | 0.8–1.1 | | 4 | 586.473 |
| | 1.2–1.3 | | 5 | 184.09 |

*3.10. Groundwater Potential Areas*

Figure 4 represents the different potential sectors through influencing factor technique (varies from poor to very high) classes in the watershed of Ain Sefra. The low potential region is found in the high mountain areas of the considered zone, which are covered by Jurassic formations and characterized by a very steep slope. A significant opportunity has arisen in the low Cretaceous regions of the study area where the infiltration is high, which is supported by the presence of gentle slope (0–3.31%). Furthermore, the concentration of drainage aids streamflow in replenishing the groundwater system.

Rainfall is crucial since it is the only source of this watershed. The high precipitation in the northern region of the study area presents a significant groundwater potential with respect to the southern regions. It has been seen that groundwater potential in the research region is influenced by geology, drainage density, and rainfall. Groundwater potential zones were identified by dividing the grids of the connecting sections into five categories, namely quite low, low, moderate, good, and excellent, which represent 0.56%, 26.41%, 44.72%, 25.22%, and 3.1% of the study area, respectively (Table 6).

**Table 6.** Groundwater recharge zones in the selected area.

| Sr. No. | Potential Zones | Area (km$^2$) | % |
|---|---|---|---|
| 1 | Very low | 25.54 | 0.56 |
| 2 | Low | 1212.27 | 26.41 |
| 3 | Moderate | 2052.55 | 44.72 |
| 4 | Good | 1157.66 | 25.22 |
| 5 | Excellent | 142.18 | 3.10 |

*3.11. Validation*

Validation is one of the most relevant scientific research criteria. For the goal of comparing the qualitative findings of the groundwater potential zones, the yield of water sources was chosen as a better option than other available data. The samples were heterogeneous to cover almost all the study area; their positions can be seen in the drainage density map (Figure 4). The highest yield among groundwater data collected was Slih (F42) with 85 L/s. While the least yield was observed from the spring of Ain Serara (S3) with an output of 2 L/s. According to the British Geological Survey, well yields can be grouped into five categories: well rates between 0 and 0.1 L/s were classified as extremely low, 0.1 and 2 L/s as low, and 2 and 5 L/s as very high. Moderate, 5 and 20 L/s, and greater than 20 L/s were

recognized as highly promising groundwater basins [75]. The classification summary of aquifer productivity of the research field is given in Table 7.

**Table 7.** Classification of aquifer productivity.

|  | Yield Range (L/s) | Class | Number of Existing Water Points | Wells Falling in the Corresponding Zones | Agreement |
|---|---|---|---|---|---|
| Very low | <0.1 | 01 | 00 | 00 | - |
| Low | 0.1–2 | 02 | 04 | 03 | 75% |
| Moderate | 2–5 | 03 | 06 | 05 | 83.33% |
| Good | 5–20 | 04 | 29 | 23 | 79.31% |
| Excellent | >20 | 05 | 06 | 04 | 66.67% |
| **Sum** |  |  | **45** | **35** | **77.78%** |

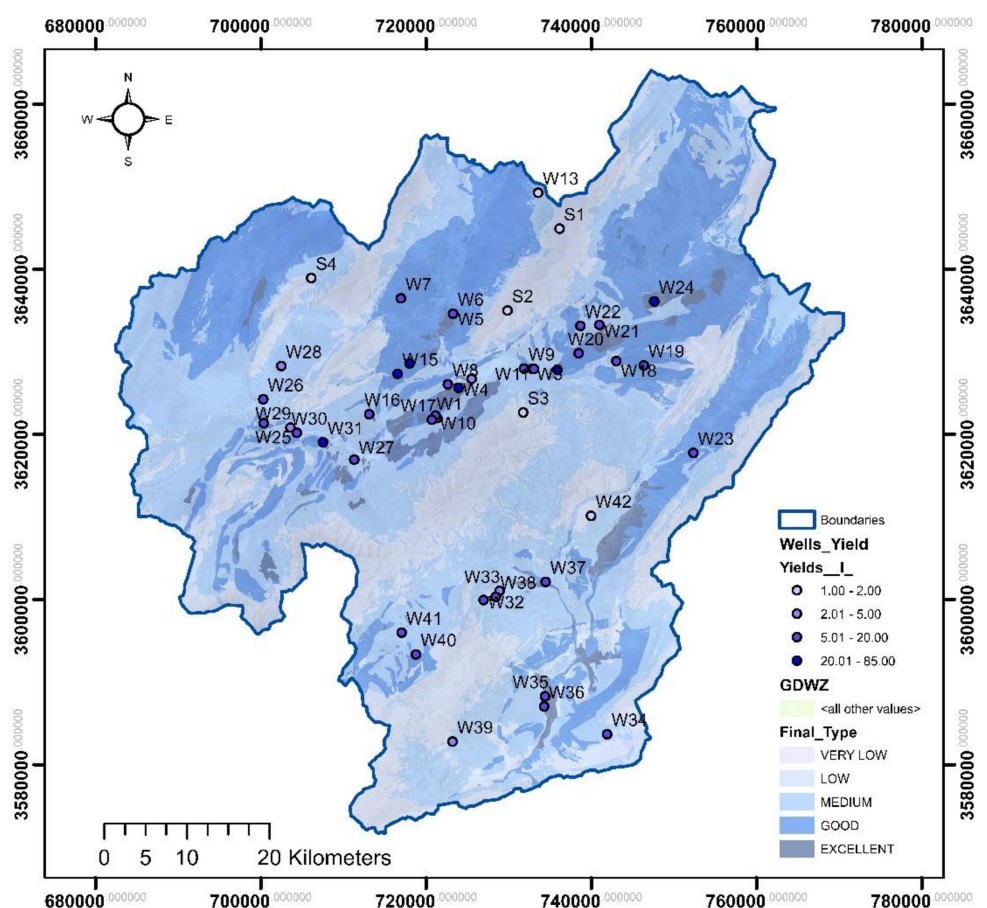

**Figure 4.** The study area's groundwater potential sectors.

According to this ranking, of the six wells with production of more than 20 L/s, four wells representing 66.67% of the collected data are identified as high groundwater recharge zones. The majority of them are reported to be low Cretaceous areas. Out of the 29 wells that supply results, 23 (79.31%) of them lie inside the excellent groundwater potential zone, whereas out of the six moderate-yield (2–5 L/s) wells, 83.33% are classified as having a moderate groundwater potential. Out of the four springs yield data, three (75%) of them fall within the area of insufficient water potential. Results provide that 77.78% of groundwater inventory data agree with the corresponding groundwater potential zone classifications. This demonstrates that the potential groundwater areas delineated using GIS and SR strategies and groundwater inventory records are in satisfactory correlation. This analysis indicates the reliability of the method used. However, some limitations should be noted:

first, the insufficient sampled boreholes, especially in the south of the study area. The second limitation concerns the lack of wells that yield between 0 and 0.1 L/s to validate the very low potentialities. Furthermore, the COVID-19 pandemic has been another challenge, especially throughout the data collection.

## 4. Conclusions

By combining remote sensing, GIS, and AHP approaches, the groundwater potential in the area of Ain Sefra, situated in the upper Wadi Namous basin in the south-western part of Algeria, was investigated. The morphometric characteristics and potential groundwater regions in the research area were effectively calculated using remote sensing data and GIS. The analytic hierarchy process was used to standardize the weight and rank of each attribute and its sub-properties that impact the watershed's groundwater potential. To integrate the systems, a total of eight layers were employed. Geology, hydrological density, precipitation, TWI, DEM, Ramp, land cover/land occupation, and realized density are some of the factors to consider. As a consequence of the maps being layered on top of one other, groundwater potential areas of the watershed were extracted. These groundwater potential areas were divided into five different categories: very low, low, medium, good, and excellent. The watershed's local GWPZ maps were created using this approach. As a result, there were 30 quantified morphometric characteristics assessed, including linear, areal, and relief. The final maps revealed that moderate groundwater potential exists in 44.72% of the basin. However, 26.41% of the watershed's potential is poor, 25.22% of the basin's potential is good, and 3.1% of the basin's potential is excellent. The five unique groundwater potential zones were then evaluated using existing groundwater inventory data, yielding roughly 77.78% agreement between the quantitative groundwater potential analysis and the groundwater inventory data. The groundwater potential map was verified using field data, indicating that this prediction method is effective and dependable. According to our findings, high groundwater potential zones are concentrated in low Cretaceous locations. The composition of the rocks, as well as the existence of a moderate slope, reinforce this conclusion, indicating a high infiltration capacity. Furthermore, the concentration of drainage aids streamflow in replenishing the groundwater system. The highland area of the reservoir, on the other hand, has relatively limited groundwater potential owing to the region's high slope and the nature of rocks with very low permeability.

We recognize that our study may have certain limitations from a lack of data, particularly in the south of the study region where direct groundwater observations are sparse, and the lack of the wells that yield between 0 and 0.1 L/s to validate the very low potentialities. Furthermore, the COVID-19 pandemic has been another challenge, especially throughout the data collection. Despite these limitations, the findings of this study are important, and it may be stated that the created groundwater potential map is a first step in locating favorable locations for new producing wells in the Ain Sefra watershed without incurring major costs and time, as well as in offering practitioners, policy makers, and stakeholders with crucial potential groundwater information that is capable of facilitating decisions and indicating research directions in terms of hydrogeological prospecting. Hence, it is recommended to install a large piezometric network in the watershed for monitoring the fluctuations of the water table to better assess the effectiveness of this method.

This approach might be used not only in the Ain Sefra region, but we believe that such assessments are crucial for groundwater development schemes in arid and semi-arid locations around the world where observed groundwater data is limited to ensure sustainable groundwater resources usage in these water-stressed regions.

**Author Contributions:** Conceptualization, A.D., A.B. and N.K.; methodology, K.M., A.M.A. and H.A. (Hijaz Ahmad); investigation, Y.M. and H.A. (Houari Ameur); Funding, K.M.; writing-original draft preparation, A.D., A.B. and N.K.; writing-review and editing, K.M., A.M.A., H.A. (Hijaz Ahmad), Y.M. and H.A. (Houari Ameur). All authors have read and agreed to the published version of the manuscript.

**Funding:** This research has received funding support from the NSRF via the Program Management Unit for Human Resources & Institutional Development, Research and Innovation [grant number B05F640204].

**Institutional Review Board Statement:** Institutional Review Board Statement and approval number are not applicable.

**Informed Consent Statement:** No informed consent statement is required for this study.

**Data Availability Statement:** All data are available in manuscript.

**Conflicts of Interest:** The authors declare no conflict of interest.

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
