# Peer review of "Groundwater Potentiality Assessment of Ain Sefra Region in Upper Wadi Namous Basin, Algeria Using Integrated Geospatial Approaches"

_sustainability, doi:10.3390/su14084450_

Round 1

Reviewer 1 Report

The study is relevant for the study region. The data are well interpreted and clearly presented. But it is necessary to edit the text: there is a lot of repetition of abbreviations and words are written meanings that can be replaced by generally accepted icons. It all makes the text bigger.

Reviewer 2 Report

The paper is well structured and elaborated. Logic is easy to follow. The only missing point that the reviewer can suggest to improve the paper is by developing the conclusion further by answering these questions:

  1. what is the limitation of the study?
  2. what is the bigger impact of the study? / what is the benefit for other regions? 
  3. arrange recommendations for: practictioners, policy makers, economic benefit, and for other stakeholders. 
